# Seroprevalence of SARS-CoV-2 Infection and Adherence to Preventive Measures in Cuenca, Ecuador, October 2020, a Cross-Sectional Study

**DOI:** 10.3390/ijerph18094657

**Published:** 2021-04-27

**Authors:** David Acurio-Páez, Bernardo Vega, Daniel Orellana, Ricardo Charry, Andrea Gómez, Michael Obimpeh, Veronique Verhoeven, Robert Colebunders

**Affiliations:** 1Faculty of Medical Science, Universidad de Cuenca, 010202 Cuenca, Ecuador; bernardo.vegac@ucuenca.edu.ec (B.V.); ricardo.charry64@ucuenca.edu.ec (R.C.); 2Grupo de Investigación LlactaLAB—Ciudades Sustentables, Universidad de Cuenca, 010203 Cuenca, Ecuador; daniel.orellana@ucuenca.edu.ec; 3School of Public Health, University of Chile, 8380453 Santiago, Chile; andrea.gomez@ug.uchile.cl; 4Family Medicine and Population Health, University of Antwerp, 2610 Antwerp, Belgium; Michael.Obimpeh@student.uantwerpen.be (M.O.); veronique.verhoeven@uantwerpen.be (V.V.); robert.colebunders@uantwerpen.be (R.C.)

**Keywords:** COVID-19, prevalence, IgM antibodies, IgG antibodies, PCR test, adherence, preventive measures, Ecuador

## Abstract

A door-to-door survey was organised in Cuenca, Ecuador, to determine the prevalence of COVID-19 infection and adherence of the population to COVID-19 preventive measures. A total of 2457 persons participated in the study; 584 (23.7%) reported having experienced at least one flu-like symptom since the onset of the pandemic. The maximum SARS-CoV-2 seroprevalence in Cuenca was 13.2% (CI: 12–14.6%) (IgM or IgG positive). Considering PCR confirmed infections, the prevalence was 11% (CI: 10–12.4%). There was no significant difference in seroprevalence between rural and urban areas. Participants aged 35–49 years old, living with a COVID-19 positive person, at least six people in a household, physical contact with someone outside the household, a contact with a person outside the home with flu-like symptoms, using public transport, and not having enough resources for living, significantly increased the odds for SARS-CoV-2 seropositivity. Overall, there was good adherence to COVID-19 preventive measures. Having known someone who tested positive for COVID-19, having a primary or secondary level of education, and having enough resources for living, significantly increased the odds for higher adherence. In conclusion, despite good overall adherence of the population of Cuenca with COVID-19 preventive measures, our study suggests high ongoing COVID-19 transmission in Cuenca, particularly in certain parishes. Prevention should not only focus on behavioural change, but on intensified testing strategies in demographical risk groups.

## 1. Introduction

With more than 62 million confirmed cases as of 30 November 2020, the Coronavirus disease 2019 (COVID-19) pandemic has severely hit all continents around the world [1,2]. Ecuador, with a population of more than 17 million people, recorded its first case of COVID-19 on the 29 February 2020 [3]. The pandemic started in the city of Guayaquil, in the Guayas province, where the health system was rapidly overwhelmed and where local funeral homes initially were incapable of handling all the deaths. A Sanitary emergency was declared at the national level on 16 March 2020, implementing several lockdown measures: restrictions of mobility, including international and domestic travel, restrictions of non-essential services, teleworking, closure of the educational system at all levels, and closure of bars and restaurants. Since 16 April, the use of face masks was mandatory for the entire population. After two months, on 4 May, a traffic-light-coded restriction system was implemented, giving the responsibility to each local government to decide the level of restriction in their jurisdiction. The restriction levels established thresholds for commercial activities, public space usage, mobility, public transport, among others. Since June, several municipalities started to relax their restriction levels, and on 12 September the National Court of Justice ruled that a further extension of the national emergency was unconstitutional. Subsequently, the restriction measures were further relaxed. By November, almost all activities except education returned to a certain normality, and general mobility reached 24% below the pre-lockdown levels [4]. Since then, mortality levels have stayed approximately 25% above the historical mean [5,6]. However, on 2 April 2021, a 30-day state of exception was declared in eight of the country’s 24 provinces, which included a curfew due to a spike in COVID-19 infections and associated deaths. On 3 April 2021, there were a total of 335,681 confirmed COVID-19 cases in Ecuador, with 16,956 deaths.

In the canton of Cuenca, in Ecuador, the first case of COVID-19 was confirmed on 14 March 2020, according to information from the Ministry of Public Health. The first symptoms started on 1 March 2020, and by 11 November 2020 there were officially 8670 confirmed cases with RT PCR and 211 deaths. However, little is known about the degree of community transmission of the virus in Cuenca, and the COVID-19 preventive behaviour of the population.

We expected a high degree of community transmission in Cuenca related to non-adherence with COVID-19 preventive measures. With this study, we aimed to determine the SARS-CoV-2 seroprevalence in Cuenca and risk factors for SARS-CoV-2 infection.

## 2. Materials and Methods

### 2.1. Study Setting and Design

This was a cross-sectional door-to-door study conducted in the canton of Cuenca, in the Azuay province, Ecuador, between 11 August and 1 November 2020. Cuenca is the third largest city of Ecuador after Guyaquil and Quito.

Sample size calculation. The Cuenca canton has a total population of 636,996 inhabitants according to the national institute of statistics; 418,152 (66%) live in an urban area and 218,844 (34%) in a rural area. The sample size was calculated in two steps. In the absence of any data to estimate the SARS CoV-2 prevalence in Cuenca, for the sample size calculation, we chose a prevalence of 50%, as the worst-case scenario, to obtain a sufficiently large sample size. In a first step, considering a SARS CoV-2 seroprevalence of 50%, a desired precision of 0.03, and a confidence level of 0.95, and considering 10% of persons not participating, in the urban areas 1220 persons and in the rural areas, 1217 persons had to be selected. In a second step, the number of participants for each parish was calculated according to the population density of the 16 urban parishes and the 21 rural parishes.

Selection participants. The municipality of Cuenca has a map of each household in the canton identified by the number of electric light contacts. This mechanism was chosen to identify houses in rural areas with unnamed streets without road access. On the cadastral map of the Cuenca, each house was numbered and randomly selected in each parish. Only points that were within properties were selected, and we linked them to the cadastral key, thus discarding points located on roads. Verification was carried out with satellite images. The selection of the households followed a quasi-random spatial sampling strategy. First, a point layer was created for each parish, with spatial-restricted random distribution with a minimum distance of 60–100 m between points and restricted to the consolidated zones to avoid inhabited areas. Then, the points were overlaid with the parish’s cadastral map to select the parcels, filtering out unoccupied, non-empty parcels. If the parcel contained more than one household, only the first one (according to the cadastral number) was selected for the sample. The selected households were stored in a new layer representing the sample’s spatial locations and used to create field maps for the surveyors. One member was selected per household by drawing lots among the members present the day of the visit of the house.

Interviews were conducted with the head of the household using a structured questionnaire. This questionnaire included questions concerning sociodemographic factors, work-related factors, education level, self-reported flu-like symptoms, history of an underlying chronic disease, and preventive behaviour in the context of COVID-19 preventive measures. To assess adherence to COVID-19 prevention measures, the ICPcovid consortium questionnaire was used in several low and middle-income countries. The complete questionnaire was pilot tested on 22 workers of the municipality of Cuenca. The COVID-19 test was carried out on a family member chosen at random among the family members present in the household. The GPS location of each household was recorded to permit revisiting households with a positive case, to counsel them about preventive measures, evaluate the health status of the infected person, and trace and test contacts.

Data collection. Data were collected on tablets or mobile phones, with the software Kobo Toolbox, and then transferred to R software for analysis.

Study approval number Ministry of Health Ecuador MSP-DNGA-SG-10-2020-7954-E.

### 2.2. Testing for COVID-19 Infection

SARS-CoV-2 seroprevalence was measured using both rapid IgM and IgG tests, the “Standard Q COVID-19 IgM/IgG Plus” (SD BIOSNSOR, Suwon, Korea Republic) [7]. The specificity and sensitivity of the test, seven days or more after the symptoms, were considered to be 94.3% and 87.9%, respectively. In the case of a positive IgM test, a PCR test was done on a nasal swab obtained at the respondents’ house, one week after the result of the IgM/IgG only on those with a positive for IgM test. A person was considered to be SARS-CoV-2 seropositive if he/she was positive for either IgM or IgG antigen test (most sensitive test). A person was considered to be SARS-CoV-2 seropositive if he/she was positive for both IgM and IgG (most specific test).

### 2.3. Assessment of Adherence

Individual adherence to the government’s restrictive instructions for COVID-19 control was assessed using individual and community parameters, configured either as yes/no questions or as Likert scales in the questionnaire. A composite adherence score was made based on the respondent’s self-reported observance of the following personal preventive measures: physical distancing, face mask use, hand hygiene, coughing hygiene, and avoidance of overcrowded places (Table 1). A dichotomised scale was generated by collapsing responses for 0 through 3 from the original scale and coded as 0 or low adherence, while responses from 4 and 5 were coded as 1 and interpreted as high adherence [8].

### 2.4. Data Processing and Analysis

Completed data were exported to a Microsoft Excel 2016 spreadsheet for cleaning and coding and subsequently transferred to R software version 4.0.3 for subsequent Statistical analysis.

Descriptive statistics were presented using means and standard deviation (SD) for continuous variables and percentages for categorical variables. Seroprevalences were calculated across categorical variables using sample proportions and 95% corresponding confidence intervals (CI).

A maximum SARS-CoV-2 seroprevalence in Cuenca was calculated by considering a SARS-CoV-2 antibody test positive if IgM or IgG positive (most sensitive test). A minimum SARS-CoV-2 seroprevalence was calculated by considering a SARS-CoV-2 antibody test positive if IgM and IgG positive (most specific test).

Bivariate regression analyses were carried out to identify variables associated with SARS-CoV-2 seropositivity (using the most sensitive test) and adherence to COVID-19 preventive measures. The variables with a likelihood ratio *p*-value < 0.25 in bivariate regression were included in the multivariate analysis. The selected variables from the bivariate analysis were subjected to a backward stepwise selection process, and a final most performant model with the least Akaike information criterion (AIC) was selected. Multi-collinearity was checked using variance inflation factors; hence the variables in the models do not depend on each other and do not render the models inaccurate in our parameter estimations. The level of significance used was 5%, and all tests were two-sided.

Multivariate analysis (using the most sensitive test) was used to investigate factors associated with SARS-CoV-2 seroprevalence and factors associated with individual adherence to national preventive measures against COVID-19. A dichotomised adherence score was used as the dependent variable to fit a logistic regression. The association between dependent and independent variables in both instances was determined using both crude odds ratios (COR) and adjusted odds ratios (AOR). A 95% confidence interval (95% CI) and *p*-value < 0.05 were used to determine the statistical significance level of the independent variables.

## 3. Results

### 3.1. Participants Characteristics

Of the 2457 persons who participated in the study, 59.4% were female, 50.1% lived in a rural area, and 49.9% in an urban area; 97.7% of them had Ecuadorian nationality, 2.3% were foreigners. The average age of participants was 39 years (SD: 19.4), with 47.3% participants over 40 years of age (Table 2).

A high SARS-CoV-2 seroprevalence was observed among the 35–49 year olds, persons in cohabitation, persons with only primary education, persons with a lower monthly income (in particular those with less than 200 $ per month), public transport (bus/taxi) users, and daily workers/house helps and small farm owners (Table 2).

### 3.2. Flu-Like Symptoms Associated with COVID-19 Infection among Participants

Overall, 23.7% of the participants reported having experienced at least one flu-like symptom since the onset of the pandemic. The most frequently reported flu-like symptoms were headache (39.6%), sore throat (34.1%), stuffy or runny nose (34.1%), cough (30%), and fatigue (27.7%) (Figure 1). Among the 272 participants who tested positive for COVID-19, 81 (30%) reported one or more flu-like symptoms, suggesting that the infection was asymptomatic in 70%. Flu-like symptoms such as headache, fever, runny nose, cough, fatigue, nausea, serious difficulty in breathing were associated with COVID-19 positivity in bivariate analysis. However, after multivariate analyses, loss of taste or smell remained the only flu-like symptom associated with COVID-19 positivity (AOR = 5.71, 3.28–9.95).

### 3.3. SARS-CoV-2 Seroprevalence in Cuenca

The maximum SARS-CoV-2 seroprevalence in Cuenca was 13.2% (CI: 12–14.6%) (IgM or IgG positive) and the minimum SARS-CoV-2 seroprevalence 4% (CI: 3.2–4.8%) (IgM and IgG positive). Considering PCR confirmed infection prevalence was 11% (CI: 10–12.4%). There was no difference in SARS-CoV-2 prevalence between rural and urban areas. However, there were important differences in seroprevalence between parishes, with some parishes having a prevalence that was 2 to 3 times higher than average. The highest prevalence was observed in Tarqui (38.8%) and Checa (36.4%) (Figure 2 and Figure 3).

Most (84%, 62/74) of the respondents in Tarqui did not have enough resources for living, in 95% (70/74) their income was less than 519 $, and 93% (69/74) lived in crowded houses. In Checa, for 80% (16/20), their income was less than 519 $, and all participants lived in houses with a minimum of 2 people in the house.

### 3.4. Factors Associated with SARS-CoV-2 Seroprevalence

A logistic regression model showed that the following factors were associated with SARS-CoV-2 seropositivity: persons in the age groups 35–49 years, a COVID-19 positive person in the home, using public transport, at least six people in a household, given a hand or a kiss in the last week with someone from outside the household, a contact with a person outside the home with flu-like symptoms, and not having enough resources for living. (Table 3).

Taking into account only COVID-19 PCR confirmed cases, the following factors were associated with COVID-19 infection: being in the age group 65 years and above, not having enough resources for living, use of public transport (bus or taxi), attendance of a funeral in the past 7 days, contact with a person with flu-like symptoms, a COVID-19 positive person in the home, given a hand or a kiss in the last week, not having enough resources for living, and more than six people living in a house (Appendix A).

### 3.5. Adherence to COVID-19 Preventive Measures

There was a high level of adherence to almost all COVID-19 preventive measures by most participants (Appendix A). Overall, 92.6% of participants reported to always respect the 1.5 m physical distancing rule when going outside and 93.2% to always wear a face mask when going outside. Only 6.8% reported never wearing a face mask. Over 90% of participants adhered to frequent hand washing, did not attend a meeting with more than 10 people, nor a religious nor funeral service, and did not travel within nor outside the country.

A logistic regression model showed that having known someone who tested positive for COVID-19, having a primary or secondary level of education, having earned a monthly income from 200 $ to 519 $, and having enough resources for living, significantly increased the odds for higher adherence. Participants who were single, being in the age group 20–34 years, earned a monthly income of at least 520 $ and 5 to 6 people in the household had reduced odds of adherence (Table 4).

A similar analyses to determine factors associated with adherence to wearing masks and keeping distance showed the same results (data not shown).

## 4. Discussion

This cross-sectional study assessed the SARS-CoV-2 seroprevalence and adherence to preventive measures in Cuenca, Ecuador. The seroprevalence was 13.2% (CI: 12–14.6%) with the most sensitive test, with no significant difference in seroprevalence between rural and urban areas. The highest seroprevalence was observed in Tarqui (38.8%) and Checa (36.4%). The lower the level of household income and the higher the number of people living in the homes, the higher the level of SARS-CoV-2 seroprevalence of the parish.

Persons in the age group 35–49 years, having a COVID-19 positive person in the home, at least six people in a household, physical contact with someone outside the household, contact with a person outside the home with flu-like symptoms, use of public transport, attendance of a funeral and not having enough resources for living were associated with COVID-19 infection. The most frequently reported flu-like symptoms related to COVID-19 were headache, sore throat, stuffy or runny nose, cough, and fatigue. Among the 272 participants who tested PCR positive for COVID-19, 81 (30%) reported one or more flu-like symptoms, suggesting that the infection was asymptomatic in 70%.

The overall level of adherence to COVID-19 preventive measures including adherence with face mask use and physical distancing was high among the participants. The latter preventive measures have been shown to be very effective in reducing COVID-19 transmission [9]. A higher level of education, with the exception of tertiary education level, was associated with a higher adherence level. This was also reported in a similar survey recently conducted in Mozambique [10]. Earning a monthly income of at least 200$ and having enough resources for living were associated with a higher adherence level. This may be because it requires resources to be adherent to the preventive measures, for example to buy and change face masks frequently and to buy hand sanitisers.

The SARS-CoV-2 seroprevalence was high even among participants who adhered to the COVID-19 preventive measures. This is unexpected because COVID-19 preventive measures have been shown to be effective in reducing transmission [11,12]. The reason may be that self-reported adherence to the preventive measures was not very reliable and that only very strict adherence to prevention measures protect against infection in case of high community transmission.

In a meta-analysis of SARS-CoV-2 seroprevalence studies, seroprevalences varied from 1.45% (0.95–1.94%, South America) to 5.27% (3.97–6.57%, Northern Europe). A nationwide seroprevalence study in Spain in April-May 2020, showed that only 5% of the population had antibodies against SARS-CoV-2 [13]. In June to July 2020, the SARS-CoV-2 seroprevalence in Lima was 20.8% (95% CI 17.2–23.5) and was higher in lower socioeconomic status and overcrowded households [14]. On June 2020, 28% of blood donors in the State of Rio de Janeiro were found to be SARS-CoV-2 seropositive [15]. However, in South America also, very high SARSCoV-2 seroprevalences have been reported. For example, in the Atahualpa project, a rural area in coastal Ecuador, 303 (45%) of 673 Ecuadorian adults were IgG/IgM SARS-CoV-2 seropositive, and 77% of them had experienced clinical manifestations of a COVID-19 infection. In this study, seropositivity was associated with the use of open latrines [16]. In Brazil, a SARS-CoV-2 seroprevalence of 40.4% (95% CI 35.6–45.3) was reported in Maranhão, and of 76% (95% CI 67–98) in October 2020 among blood donors in Manaus [17]. High attack rates of SARS-CoV-2 were also estimated in population-based samples from other locations in the Amazon Basin—e.g., Iquitos, Peru (70%, 67–73) [18].

On 21 January 2021, COVID-19 vaccination started in Ecuador [19], and on 26 March 2021, 191,179 health care workers and the elderly had already received at least a first dose of the Pfizer vaccine. However, it will take many months before enough people are vaccinated to stop COVID-19 community transmission. Therefore currently, preventive measures such as mask-wearing, frequent hand washing, and observing the minimum 1.5 physical distancing rule should not be relaxed [20,21,22].

However, since in our sample transmission was high, even in people with self-reported good adherence to preventive measures, focusing mainly or solely on behavioural change does not seem to be sufficient. A complementary strategy would be to increase testing in demographical risk groups, for example, in highly affected parishes. Furthermore, testing could be offered in public places such as public transport terminals, at the beginning of funerals and church services, and testing could be encouraged in large households and in certain professions such as daily workers, house helps, and farmers.

## 5. Limitations of the Study

The strength of our study is that we carried out a door-to-door survey of a representative sample of the population of Cuenca and that COVID-19 IgM test results were confirmed with a PCR result. The COVID-19 IgM/IgG test only becomes positive one week after infection. Therefore, certain individuals may have been only recently infected and not yet be COVID-19 IgM/IgG positive. For assessing factors associated with COVID-19 infection, it would have been better if PCR testing had been done in all participants to pick up recent infections. This means our study underestimates the degree of SARS-CoV-2 infection in Cuenca. A limitation of our study is that we did not adjust the estimated prevalence for the sensitivity and specificity of the test. Moreover, the adherence to preventive measures was only assessed through interviews, and therefore, socially desirable answers may have been given by the study participants. Finally, we only carried out a limited assessment of the socioeconomic status of the participants and we did not register the non-response rate.

## 6. Conclusions

Despite good overall adherence with COVID-19 preventive measures, our study suggests high ongoing COVID-19 transmission in Cuenca, particularly in certain parishes. Similar to other studies, a lower socioeconomic status, and overcrowded households were found to be risk factors for SARS-CoV-2 infection. A large-scale COVID-19 vaccination will be needed to stop the pandemic in Cuenca.

## Figures and Tables

**Figure 1 ijerph-18-04657-f001:**
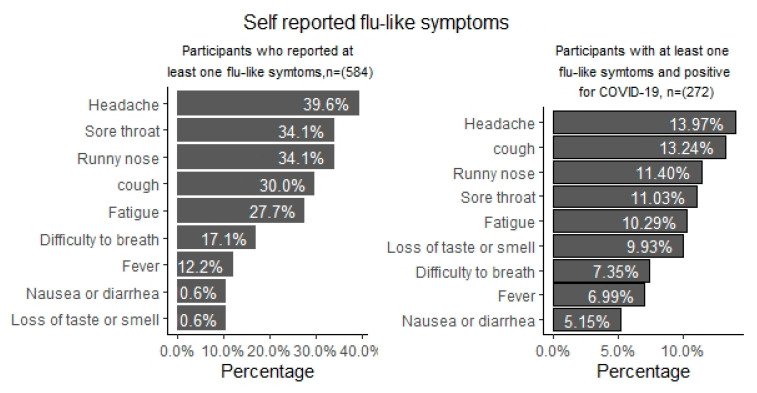
Self-reported flu-like symptoms among participants in Ecuador.

**Figure 2 ijerph-18-04657-f002:**
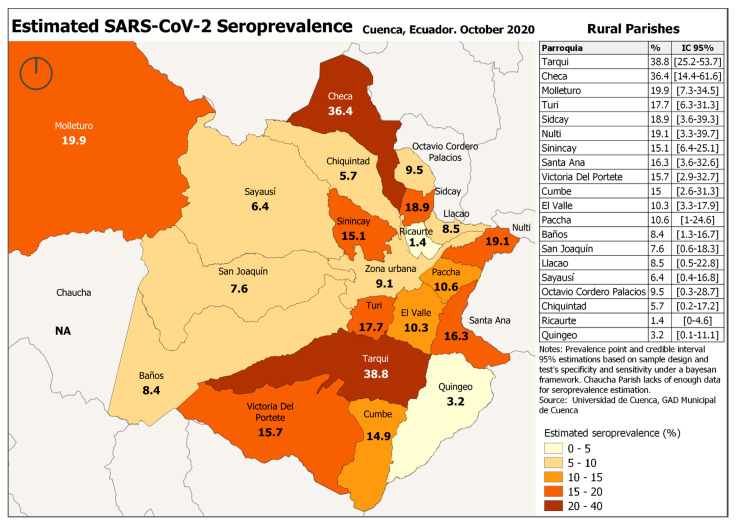
Estimated SARS-CoV-2 Seroprevalence, Cuenca, Ecuador 2020 Rural parishes.

**Figure 3 ijerph-18-04657-f003:**
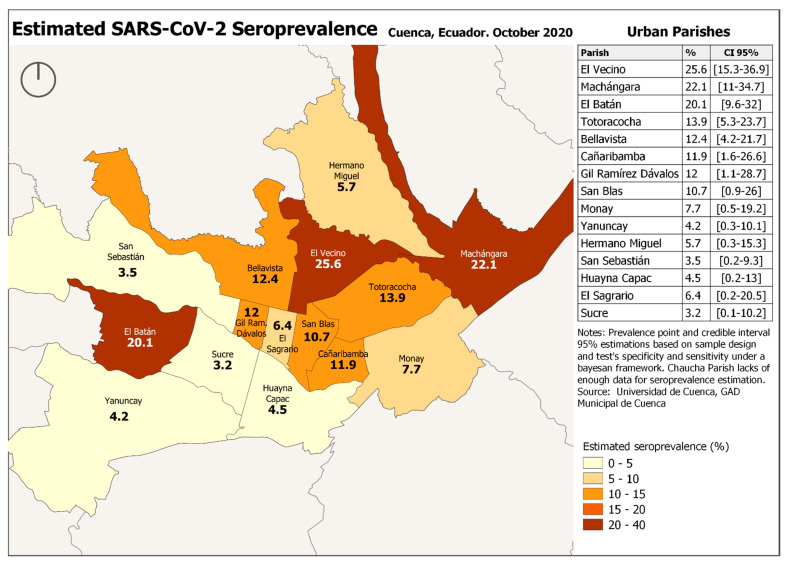
Estimated SARS-CoV-2 Seroprevalence, Cuenca, Ecuador 2020 Urban parishes.

**Table 1 ijerph-18-04657-t001:** Composition of the adherence score to COVID-19 preventive measures.

Variable	Scoring	Interpretation	
I follow the 1.5–2 m physical distance rule	Yes no	1 0	1 point for yes, 0 points for no
I wash my hands regularly or I use hand sanitiser regularly	Yes No	1 0	If any (or both) questions have answered yes, 1 point, 0 points for no
I stay at home when I have flu symptoms	Yes no	1 0	1 point for yes, 0 points for no
I wear a face mask when I go out	Yes no	1 0	1 point for yes, 0 points for no
I respect not going out to overcrowded places	Yes no	1 0	1 point for yes, 0 points for no
Total adherence score (maximum): 5			

**Table 2 ijerph-18-04657-t002:** SARS-CoV-2 seroprevalence by general characteristics of the participants.

Variables	Response	No. of Positive/No. of Subjects	SARS-CoV-2 Seroprevalence (95% CI)
Sex	Male, *n* (%)	118/997	12% (10–14)
	Female, *n* (%)	207/1460	14.2% (12.5–16.1)
Age group	0–19 years, *n* (%)	45/402	11.2% (8.4–14.8)
	20–34 years, *n* (%)	79/675	11.7% (9.4–14.3)
	35–49 years, *n* (%)	100/624	16% (13.3–19.2)
	50–64 years, *n* (%)	63/461	13.7% (10.7–17.2)
	65+ years, *n* (%)	38/295	13% (9.3–17.4)
Parish	Rural, *n* (%)	168/1231	13.6% (11.8–15.7)
	Urban, *n* (%)	157/1226	12.8% (11–14.8)
Marital status	Divorced, *n* (%)	27/193	14% (9.5–20)
	Single, *n* (%)	95/797	12% (9.8–14.4)
	Married, *n* (%)	155/1122	13.8% (12–16)
	Cohabitation, *n* (%)	30/186	16.1% (11.3–22.4)
	Widower, *n* (%)	11/96	11.5% (6.1–20)
Education	None, *n* (%)	15/103	14.6% (8.6–23)
	Primary, *n* (%)	148/859	17.2% (14.8–20)
	Secondary, *n* (%)	116/883	13.1% (11–15.6)
	Tertiary, *n* (%)	45/602	7.5% (5.5–10)
Monthly income	Less than 200 $, *n* (%)	108/682	15.8% (13.2–18.8)
	From 200 $ to 519 $, *n* (%)	144/1052	13.7% (11.7–16)
	From 520 $ to 1500 $, *n* (%)	61/531	11.5% (9–14.5)
	More than 1500 $, *n* (%)	7/95	7.4% (3.3–15.1)
Nationality	Ecuador, *n* (%)	319/2400	13.3% (12–14.7)
	Foreigner, *n* (%)	6/57	10.5% (4.3–22)
Housing conditions	House with Garden, *n* (%)	145/1166	12.4% (10.6–14.5)
	House without Garden *n* (%)	140/980	14.3% (12.2–16.7)
	Apartment, *n* (%)	23/202	11.4% (7.5–16.7)
	Quarters, *n* (%)	14/96	14.6% (8.5–23.6)
	Cabin, *n* (%)	3/13	23.1% (6.2–54)
Means of transport to work	On Foot, *n* (%)	65/529	12.6% (9.6–15.5)
	Bicycle/Moto/Trolley car, *n* (%)	12/106	11.3% (6.2–19.3)
	Private car, *n* (%)	99/912	11% (9–13.1)
	Public (Bus/Taxi), *n* (%)	135/742	18.2% (15.5–21.2)
Occupation	None, *n* (%)	66/610	10.8% (8.5–13.6)
	Daily workers/House help, *n* (%)	39/197	20% (14.6–26.2)
	Private business owners, *n* (%)	134/1102	12.2% (10.3–14.3)
	Street sellers, *n* (%)	4/31	13% (4.2–30.8)
	Small farm owners, *n* (%)	26/85	30.6% (21.3–41.7)
	Pensioners, *n* (%)	10/121	8.3% (4.3–15.1)
	Others, *n* (%)	43/214	20% (15.1–26.2)

**Table 3 ijerph-18-04657-t003:** Risk factors associated with COVID-19 IgM/IgG seroprevalence.

Covariates	Crude OR (95% CI)	Adjusted OR (95% CI)	*p*-Value
Gender			
Male	Ref	Ref	
Female	1.24 (0.95–1.56)	1.12 (0.88–1.46)	0.190
Age			
0–19 years,	Ref	Ref	
20–34 years,	1.10 (0.74–1.66)	1.07 (0.69–1.64)	0.751
35–49 years,	1.53 (1.04–2.27)	1.56 (1.07–2.46)	0.023
50–64 years,	1.34 (0.83–2.06)	1.38 (0.96–2.35)	0.078
65+ years,	1.36 (0.84–2.21)	1.45 (0.87–2.43)	0.153
Enough resources for living			
No	Ref	Ref	
Yes	0.65 (0.51–0.84)	0.64 (0.49–0.84)	0.001
Means of transport			
Private (own car, foot, bicycle)	Ref	Ref	
Public (bus/taxi)	1.73 (1.36–2.21)	1.65 (1.28–2.14)	<0.001
Has anyone in your house had COVID-19?			
No	Ref	Ref	
Yes	5.91 (4.01–8.7)	6.21 (4.12–9.37)	<0.001
Being in contact with someone outside your home with flu-like symptoms,			
No	Ref	Ref	
Yes	1.90 (1.40–2.57)	1.83 (1.32–2.53)	<0.001
No of people in house			
Living alone	Ref	Ref	
2–4 people	1.08 (0.72–1.61)	1.07 (0.71–1.63)	0.739
5–6 people	1.05 (0.68–1.63)	1.02 (0.64–1.6)	0.945
>6 people	2.41 (1.47–3.95)	2.21 (1.3–3.75)	0.003
Physical contact * with someone outside the household			
No	Ref	Ref	
Yes	1.49 (1.17–1.89)	1.54 (1.20–1.98)	<0.001

* given a hand or kiss.

**Table 4 ijerph-18-04657-t004:** Factors associated with adherence to COVID-19 preventive measures.

Covariates	Frequency (%)		Crude OR (95% CI)	Adjusted OR (95% CI)	*p*-Value
	Non Adherence	High Adherence			
Gender					
Male	117 (11.7)	880 (88.3)	Ref	Ref	
Female	122 (8.36)	1338 (91.6)	1.45 (1.11–1.9)	1.28 (0.95–1.74)	0.107
Age					
0–19 years,	39 (9.7)	363 (90.3)	Ref	Ref	
20–34 years,	88 (13)	587 (87)	0.71 (0.47–1.08)	0.56 (0.35–0.9)	0.017
35–49 years,	53 (8.5)	571 (91.5)	1.18 (0.76–1.85)	0.96 (0.58–1.58)	0.872
50–64 years,	36 (7.8)	425 (92.2)	1.25(0.77–2.03)	0.84 (0.49–1.43)	0.518
65+ years,	23 (7.8)	272 (92.2)	1.22 (0.71–2.11)	0.99 (0.54–1.84)	0.993
Do you know anyone who has Corona					
No	195 (10.7)	1619 (89.3)	Ref	Ref	
Yes	44 (6.8)	599 (93.2)	1.78 (1.26–2.53)	1.75 (1.2–2.54)	0.004
Education					
None	26 (25.2)	77 (74.8)	Ref	Ref	
Primary	46 (5.4)	813 (94.6)	6.3 (3.66–10.85)	5.31(2.86–9.88)	<0.001
Secondary	70 (8)	813 (92)	4.16 (2.5–6.95)	3.59 (1.94–6.65)	<0.001
Tertiary	97 (16)	505 (84)	1.82 (1.11–3)	1.59 (0.84–3.01)	0.154
Monthly income					
Less than 200$	64 (9.4)	618 (90.6)	Ref	Ref	
From 200$ to 519$	87 (8.3)	965 (91.7)	1.15 (0.82–1.61)	0.6 (0.4–0.9)	0.014
From 520$ to 1500$	69 (13)	462 (87)	0.69 (0.48–0.99)	0.43 (0.26–0.68)	<0.001
More than 1500$	14 (14.7)	81 (85.3)	0.6 (0.32–1.12)	0.25 (0.12–0.54)	<0.001
Enough resources for living					
No	157 (20)	629 (80)	Ref	Ref	
Yes	82 (5)	1589 (95)	5.08 (3.82–6.77)	7.14 (5.09–10)	<0.001
Number of people in house					
Living alone	18 (6)	276 (94)	Ref	Ref	
2–4 people	94 (7.2)	1214 (92.8)	0.87 (0.52–1.47)	0.79 (0.45–1.38)	0.413
5–6 people	115 (17.4)	547 (82.6)	0.33 (0.19–0.55)	0.35 (0.2–0.61)	<0.001
>6 people	12 (6.2)	181 (93.8)	1.11 (0.51–2.4)	1.04 (0.46–2.36)	0.931

## Data Availability

The data presented in this paper are available upon reasonable request to the corresponding author.

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
