# Peer review of "Seroprevalence of SARS-CoV-2 Infection and Adherence to Preventive Measures in Cuenca, Ecuador, October 2020, a Cross-Sectional Study"

_ijerph, 2021, doi:10.3390/ijerph18094657_

Round 1
Reviewer 1 Report
- Add values:% and n in the abstract;
- Add aims or hypotheses;
- Why was the age group divided starting at 30 ?;
- How did illiterate participants manage to respond?
- What questions made up the research form?
- -Was there a validation or pre-test?
- If so, how did the instrument's validation process take place (content, clarity and objectivity) and what technique was used for consensus in the questions. This is very important since most of these questions have instruments validated in the literature for this subject. In the absence of a validation process, the research is subject to measurement and interpretation visions.
Author Response
Response to reviewer 1
Point 1
Add values:% and in the abstract;
Response
We now added numbers and % in the abstract
Point 2 Add aims or hypotheses;
Response
“With this study we aimed to determine the SARS-CoV-2 seroprevalence in Cuenca and risk factors for SARS-CoV-2 infection.”
The hypothesis was that we expected a high degree of community transmission in Cuenca and that this was related to non-adherence with COVID-19 preventive measures.
We now added at the end of the introduction “We expected a high degree of community transmission in Cuenca related to non-adherence with COVID-19 preventive measures.
Point 3. Why was the age group divided starting at 30 ?;
Response
Point 4. How did illiterate participants manage to respond?
Response
All participants were interviewed by the research team.
Point 5. What questions made up the research form?
Response
The questionnaire included questions concerning sociodemographic factors, work-related factors, education level, self-reported flu-like symptoms, history of an underlying chronic disease and preventive behaviour in the context of COVID-19 preventive measures.
Point 6. Was there a validation or pre-test?
Response
The questionnaire was piloted tested in 22 workers of the municipality of Cuenca.
Point 7. If so, how did the instrument's validation process take place (content, clarity and objectivity) and what technique was used for consensus in the questions. This is very important since most of these questions have instruments validated in the literature for this subject. In the absence of a validation process, the research is subject to measurement and interpretation visions
Response
To assess adherence to COVID-19 prevention measures, the ICPcovid consortium questionnaire was used that had been tested in many Low and Middle Income Countries. The complete questionnaire was piloted tested in 22 workers of the municipality of Cuenca.

Reviewer 2 Report
Nice manuscript. A couple of suggestions:
- The authors should update epidemiological information (both global and local) in the first paragraph
- Why did the authors assume a 50% SARS-CoV-2 prevalence for sample size estimation? The assumed prevalence was very high
- It is confusing how they came upon the selected number of households; in the first paragraph of the methods, they mentioned the total population size, not the total number of households
- Which formula or sample size calculator did the authors use?
- Although residents in urban areas comprise 66% of the population, numbers of urban and rural households were equal in the final sample. Why?
- The authors said that they selected households in each parish randomly. They failed to explain how they did random sampling. Their methodology needs much clarification at this point
- Again, the authors should explain how they randomly selected a member of a given household
- In terms of definition of seropositives, why did the authors use the sensitive definition in the multivariable analyses?
- The reported sensitivity and specificity of the antibody tests used were suboptimal. The authors should adjust the estimated prevalence for the sensitivity and specificty of the test they used
- What definition of seroprevalence, sensitive or specific, is used in table 3?
- I believe that it would be more useful if the authors studied factors of adherence to the two most important measures, ie wearing masks and keeping physical distance, instead of collapsing many measures in a simple dichotomous var
- Why certain areas had very high prevalence, >30%?
- The discussion is very poor; the authors should compare their seroprevalence data and the factors associated with them with other studies in the literature
Author Response
Response to reviewer 2
Nice manuscript. A couple of suggestions:
1.The authors should update epidemiological information (both global and local) in the first paragraph
Response
We now updated the epidemiological data until April 5nd
2.Why did the authors assume a 50% SARS-CoV-2 prevalence for sample size estimation? The assumed prevalence was very high
Response
In the absence of any data to estimate the SARS CoV-2 prevalence in Cuenca, for the sample size calculation we chose a prevalence of 50%, as the worst case scenario, to obtain a sufficiently large sample size.
We agree, retrospectively this is a high percentage. However 50% was used because of the high COVID-19 related morbidity and mortality at the start of the pandemic in Ecuador and some other South American countries such as Brazil.
3.It is confusing how they came upon the selected number of households; in the first paragraph of the methods, they mentioned the total population size, not the total number of households
Response
We now replaced in the first paragraph of the methods household by persons.
Moreover we changed the subtitle: “selection of households and participants” to “selection of participants”
We now state “In a first step, considering an SARS CoV-2 seroprevalence of 50% a desired precision of 0.03, and a confidence level of 0.95 and considering 10% of persons not participating, in the urban areas 1220 persons and in the rural areas, 1217 persons had to be selected;”
In the text we explain that per household one participants was chosen at random.
4.Which formula or sample size calculator did the authors use?
For the calculation of the sample size, the following formula was used:
- N: Total population
- Z Value, corresponding to significance (95%) Z1-a = 1,96
- Proportion estimated at the time of the survey. p = (50%) = 0,50.
- degree of precision (97%) d = 3%
This formula was used for each zone (urban and rural).
5.Although residents in urban areas comprise 66% of the population, numbers of urban and rural households were equal in the final sample. Why?
Response
The result with the formula for finite populations obtain that value for each zone.
For calculating the sample size, we used a two-step procedure in which we calculated numbers for urban and rural areas separately. Given the fact that the population is large (>100000 in both groups) the calculated sample size is not significantly different in both groups. In retrospect, it would have been better to calculate a sample size for the total population and stratify afterwards (as we did with the parishes). However, since we were very (too) conservative in our parameter settings (we over-sampled), the global sample is still more than sufficient to draw conclusions.
6.The authors said that they selected households in each parish randomly. They failed to explain how they did random sampling. Their methodology needs much clarification at this point
Response
We now clarify the way the households were chosen.
We now state “On the cadastral map of the Cuenca, each house was numbered and randomly selected in each parish. Only points that were within properties were selected and we linked them to the cadastral key, thus discarding points located on roads. Verification was carried out with satellite images. The selection of the households followed a quasi-random spatial sampling strategy. First, a point layer was created for each parish, with spatial-restricted random distribution with a minimum distance of 60m - 100m between points and restricted to the consolidated zones to avoid inhabited areas. Then, the points were overlaid with the parish's cadastral map to select the parcels, filtering out unoccupied, non-empty parcels. If the parcel contained more than one household, only the first one (according to the cadastral number) was selected for the sample. The selected households were stored in a new layer representing the sample's spatial locations and used to create field maps for the surveyors. One member was selected per household by drawing lots among the members present the day of the visit of the house.
7.Again, the authors should explain how they randomly selected a member of a given household
Response
Participants were selected by the drawing of lots among members of the house.
8.In terms of definition of seropositives, why did the authors use the sensitive definition in the multivariable analyses?
Response
We used the most sensitive definition better detect an association with risk factors
9.The reported sensitivity and specificity of the antibody tests used were suboptimal. The authors should adjust the estimated prevalence for the sensitivity and specificity of the test they used
Response
Instead of using the sensitivity and specificity of the test we used the following three criteria to determine the prevalence of infection. A person was considered to be SARS-CoV-2 seropositive if he/she was positive for either IgM or IgG antigen test (most sensitive test). A person was considered to be SARS-CoV-2 seropositive if he/she was positive for both IgM and IgG (most specific test). We also calculated the prevalence based on PCR confirmed cases.
The maximum SARS-CoV-2 seroprevalence in Cuenca was 13.2% (CI: 12% - 14.6%)(IgM or IgG positive) and the minimum SARS-CoV-2 seroprevalence 4% (CI: 3.2% - 4.8%) (IgM and IgG positive). Considering PCR confirmed infections the prevalence was 11% (CI: 10%- 12.4%).
As a limitation of the study we now mention that we did not adjust the estimated prevalences for the sensitivity and specificity of the test.
10.What definition of seroprevalence, sensitive or specific, is used in table 3?
Response
The most sensitive test
11.I believe that it would be more useful if the authors studied factors of adherence to the two most important measures, ie wearing masks and keeping physical distance, instead of collapsing many measures in a simple dichotomous var
Response
We also analyzed the factors associated with adherence to wearing masks and keeping distance. Results were similar with the analysis taking into account the 5 preventive measures. We now mention this in the text.
12.Why certain areas had very high prevalence, >30%?
Response
High prevalence was most likely associated with poverty and living in an overcrowded setting.
We now added in the text “The highest prevalence was observed in Tarqui (38.8%) and Checa (36.4%) (Figure 1 and 3). Most, 84% (62/74) of the respondents in Tarqui did not have enough resources for living, in 95 % (70/74) their income was less than 519$ and 93% (69/74) lived in crowded houses. In Checa, 80% (16/20) their income was less than 519$, and all participants lived in houses with a minimum of 2 people in the house.”
13.The discussion is very poor; the authors should compare their seroprevalence data and the factors associated with them with other studies in the literature
Response
We now compare the seroprevalence observed in Cuenca with other seroprevalence studies in South America and added several references.
We added in the discussion the following text “In a meta-analysis of SARS-CoV-2 seroprevalence studies, seroprevalences varied from 1.45% (0.95–1.94%, South America) to 5.27% (3.97–6.57%, Northern Europe). A nationwide seroprevalence study in Spain in April-May 2020, showed that only 5% of the population had antibodies against SARS-CoV-2. In June-July 2020 the SARS-CoV-2 seroprevalence in Lima was 20·8% (95% CI 17·2–23·5) and was higher in lower socioeconomic status and overcrowded households. In June 2020, 28% of blood donors in the State of Rio de Janeiro were found to be SARS-CoV-2 seropositive. However, in South America also very high SARSCoV-2 seroprevalences have been reported. For example in the Atahualpa project, a rural area in coastal Ecuador, 303 (45%) of 673 Ecuadorian adults were IgG/IgM SARS-CoV-2 seropositive, and 77% of them had experienced clinical manifestations of a COVID-19 infection. In this study seropositivity was associated with the use of open latrines. In Brazil, a SARS-CoV-2 seroprevalence of 40.4% (95%CI 35.6-45.3) was reported in Maranhão, and of 76% (95% CI 67–98) in October 2020 among blood donors in Manaus. High attack rates of SARS-CoV-2 were also estimated in population-based samples from other locations in the Amazon Basin—e.g., Iquitos, Peru 70% (67–73).”
Reviewer 3 Report
Introduction is very short - some more information on geospatial differences in terms of policy or key outbreaks would be useful. Also cultural background which differentiates Columbia from other countries (general government perception) would also be of interest to readers.
Results: Please add more summarised results of the tables in the text in terms of key findings, allowing the movement of tables to the SI and for easier reading and interpretation by the readers.
Discussion: Incredibly short and lacking a lot of detail on the findings. I expected to see context of the results in comparison to other countries globally (in line with the journal) and more information on whether any key differences or novel findings were observed relative to previous studies. The discussion reads more like the results and should be rewritten and at least doubled in length.
Limitations: Also far too short. Listing strengths in the limitations is not appropriate when the limitations are themselves so short and poorly described. I expect a much more thorough description of potential issues e.g. misrepresentation of the results, biases in study design, issues with testing etc.
Conclusion: This conclusion is too brief and does not add to the scientific literature. Please provide more critical findings pertaining to the study in a global context.
Figures: Please increase the resolution and quality. The images are compressed and look like copy/paste (Very low quality)
Tables: Please Move descriptive tables to the SI (Table 4 onwards are very large and difficult to read). Fewer tables in the main manuscript would make it easier for the audience to read and interpret with key findings in text only.
Author Response
Response to reviewer 3
Point 1. Introduction is very short - some more information on geospatial differences in terms of policy or key outbreaks would be useful. Also cultural background which differentiates Columbia from other countries (general government perception) would also be of interest to readers.
Response
We now updated the COVID-19 situation in Ecuador and included additional information in the introduction about the COVID-19 epidemiological situation in Ecuador
We added in the introduction “Ecuador with a population of more than 17 million people recorded its first case of COVID-19 on the 29th of February 2020 [3]. The pandemic started in the city of Guayaquil, in the Guayas province where the health system was rapidly overwhelmed and where local funeral homes initially were incapable to handle all the deaths.”
And
“However April 2nd, 2021 a 30-day state of exception was declared in eight of the country’s 24 provinces, which included a curfew, due to a spike in COVID-19 infections and associated deaths. April 3th 2021, there were a total of 335,681 confirmed COVID-19 cases in Ecuador with 16,956 deaths.”
Point 2. Results: Please add more summarised results of the tables in the text in terms of key findings, allowing the movement of tables to the SI and for easier reading and interpretation by the readers.
Response
We omitted Table 2 and included more text between tables to highlight key findings
Point 3. Discussion: Incredibly short and lacking a lot of detail on the findings. I expected to see context of the results in comparison to other countries globally (in line with the journal) and more information on whether any key differences or novel findings were observed relative to previous studies. The discussion reads more like the results and should be rewritten and at least doubled in length.
Response
We now compare the Cuenca data with seroprevalence data from other countries
“In a meta-analysis of SARS-CoV-2 seroprevalence studies, seroprevalences varied from 1.45% (0.95–1.94%, South America) to 5.27% (3.97–6.57%, Northern Europe). A nationwide seroprevalence study in Spain in April-May 2020, showed that only 5% of the population had antibodies against SARS-CoV-2. In June-July 2020 the SARS-CoV-2 seroprevalence in Lima was 20·8% (95% CI 17·2–23·5) and was higher in lower socioeconomic status and overcrowded households. In June 2020, 28% of blood donors in the State of Rio de Janeiro were found to be SARS-CoV-2 seropositive. However, in South America also very high SARSCoV-2 seroprevalences have been reported. For example in the Atahualpa project, a rural area in coastal Ecuador, 303 (45%) of 673 Ecuadorian adults were IgG/IgM SARS-CoV-2 seropositive, and 77% of them had experienced clinical manifestations of a COVID-19 infection. In this study seropositivity was associated with the use of open latrines. In Brazil, a SARS-CoV-2 seroprevalence of 40.4% (95%CI 35.6-45.3) was reported in Maranhão, and of 76% (95% CI 67–98) in October 2020 among blood donors in Manaus. High attack rates of SARS-CoV-2 were also estimated in population-based samples from other locations in the Amazon Basin—e.g., Iquitos, Peru 70% (67–73).”
Point 4. Limitations: Also far too short. Listing strengths in the limitations is not appropriate when the limitations are themselves so short and poorly described. I expect a much more thorough description of potential issues e.g. misrepresentation of the results, biases in study design, issues with testing etc.
Response
We now added as a limitation: “A limitation of our study is that we did not adjust the estimated prevalences for the sensitivity and specificity of the test. Moreover, the adherence to preventive measures was only assessed through interview and therefore society desirable answers may have been given by the study participants. Finally, we only did a limited assessment of the socio-economic status of the participants. Finally, we did not register the non-response rate.”
Point 5. Conclusion: This conclusion is too brief and does not add to the scientific literature. Please provide more critical findings pertaining to the study in a global context.
We now added in the conclusion” Similar to other studies a lower socioeconomic status and overcrowded households were found to be risk factors for SARS-CoV-2 infection.”
Response
Point 6. Figures: Please increase the resolution and quality. The images are compressed and look like copy/paste (Very low quality)
Response
We now included a Figure of better quality
Point 7. Tables: Please Move descriptive tables to the SI (Table 4 onwards are very large and difficult to read). Fewer tables in the main manuscript would make it easier for the audience to read and interpret with key findings in text only
Response
We propose to keep Table 4 and Table 6 but made Table 5 a supplementary table. We also added a second supplementary STable 2. Risk Factors associated with COVID-19 positivity confirmed by PCR test.
Round 2
Reviewer 1 Report
The authors responded to all suggestions
Reviewer 2 Report
The authors addressed most of my comments